# FaLA: Fast Linear Adaptation for Replacing Backbone Models on Edge Devices

**Shuo Huang, Lizhen Qu\*, Xingliang Yuan, Chunyang Chen**
Faculty of Information and Technology, Monash University
{shuo.huang1, lizhen.qu, xingliang.yuan, chunyang.chen}@monash.edu

## Abstract

In this work, we study the language model backbone replacement problem for personalized downstream tasks in a non-stationary on-device scenario. In real world, company may periodically update the knowledge and architectures of backbones to keep the competitive in the market, meanwhile, to accommodate the users' own preference, models are personalized to fit users' own distribution locally. Traditional full model tuning or transfer learning for such replacements often incur considerable local device training costs and necessitate extensive backpropagation within deep transformer layers. Addressing this issue, we propose a novel, lightweight tuning method for personalized NLP classification tasks post-backbone replacement. Our approach leverages a personalized matrix calculated from documents corresponding to users' old and new backbones. This matrix facilitates top-layer parameter tuning, drastically reducing backpropagation computation. To further mitigate training costs associated with matrix linear optimization, we employ correlation clustering to curate a few examples from personalized cluster sets for individuals. Our method achieves over 1000 times computation reduction in Flops for backpropagation and brings the user-specific initialization for personal matrix yielding significant performance boost compared with popular transfer learning methods.

## 1 Introduction

Current Natural Language Processing (NLP) models, including the ones developed for edge devices (Vucetic et al., 2022), heavily rely on pretrained backbone models, such as BERT (Devlin et al., 2018) and RoBERTa (Liu et al., 2019). Recent studies show that it is important to personalize models on edge devices because of the privacy concerns and the distribution discrepancies between local data and public training data (Yan et al., 2022).

Due to the fast growing popularity of large language models (LLMs), novel pre-trained models are released almost every week. However, the computing power of edge devices is limited. How can edge devices benefit from the most recent advances of pre-trained backbone models without incurring regular and high training costs in local devices? Fig. 1 illustrates when and how the replacement of backbone models may occur. For example, at an early stage, an on-device NLP model, consisting of Glove embeddings (Goldberg and Levy, 2014) and the task-specific layers, was fine-tuned using the private data on a local device. Thus, the model may well remember personal information and adapt itself well into the local distribution. When a new pre-trained model is released, such as BERT , the user is eager to use the new backbone model but does not want to fine-tune the whole model again on the on-device data after the replacement.

Parameter-efficient tuning (Zhuang et al., 2023) and continual learning (De Lange et al., 2021) are the closest research areas to address this problem. The current parameter-efficient tuning techniques need to update only a small proportion of model parameters during training, but they still need to run expensive backpropagation throughout the whole models. Continual learning algorithms focus on mitigating catastrophic forgetting, while learning new tasks, but they do not consider changing backbone models with low costs.

Therefore, we propose a *novel* task: efficient personalized model tuning after replacing backbone models and provide the *first* solution, Fast Linear Adaptation(FaLA), that avoids backpropagation through the backbone models after replacement. Herein, we keep the logits of the old model on the private training data and apply forward propagation of the new backbone model to obtain hidden representations for the task-specific layers. The new hidden representations are used to learn a personal matrix that is regularized by the logits of the

---

\*Corresponding author.

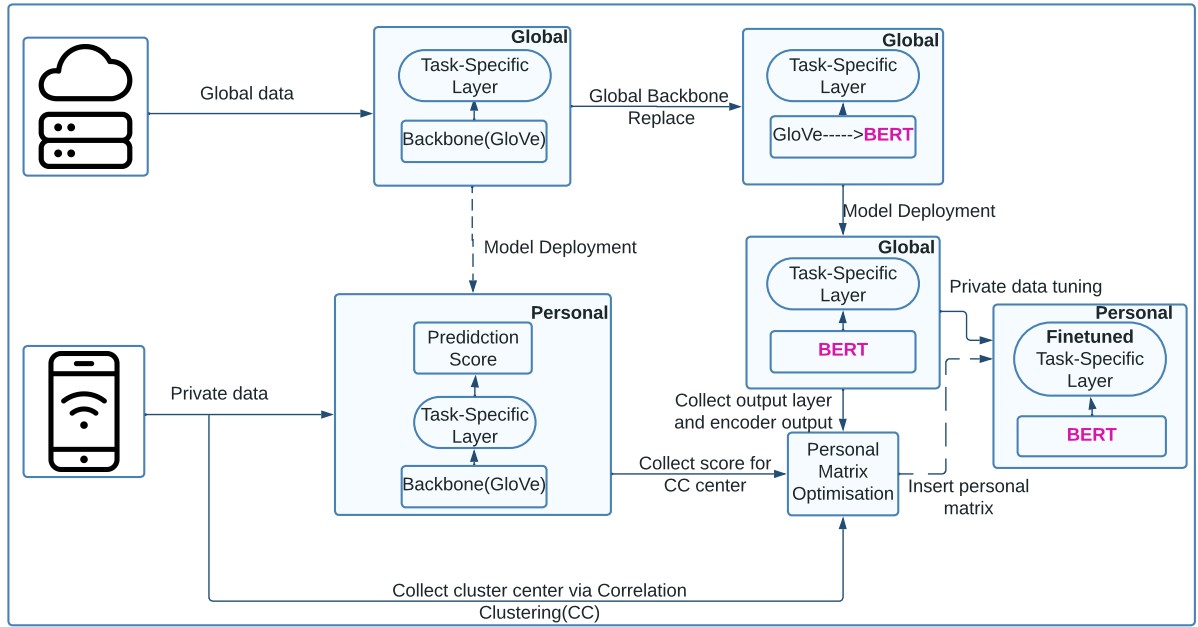

Figure 1: The replacement scenario and workflow of FaLA .

old model. The corresponding learning problem becomes a simple linear regression problem that is feasible for resource limited devices. In addition, we show that the tuning speed can be further improved by applying correlation clustering (Saha and Subramanian, 2019) and sample representative training examples from the clusters. Through extensive experiments, we show that

- Our method significantly outperforms the competitive baseline that only fine tunes the parameters of task specific layers, in terms of F1 on two tasks: News Recommendation (Wu et al., 2022) and Hate Speech Detection (Kanclerz et al., 2022).

- Our method consumes only 5.45e+9 FLOPS on News Recommendation , which is several magnitudes lower than that of Adapter (1.40e+12) (He et al., 2021) and LoRA (1.35e+12) (Hu et al., 2021) using the same backbone models.

## 2 Related Work

**On-device Personalization** Personalization refers to the incorporation of individual characteristics and user-dependent contextual information into generally trained models. Contrary to the assumption that globally trained models interpret texts with the same meaning for everyone, personalization

techniques (Flek, 2020) offer personal-oriented predictions under user-specific contexts.

As the data used for personalization often associates with sensitive information like gender, age, or user history behaviors, one compelling approach is to conduct personalization locally especially under the regulations like GDPR (Voigt and Von dem Bussche, 2017). Wang (Wang et al., 2019) evaluated on-device personalization under a federated learning setting where personal data is preserved on local devices. Their experiment on virtual keyboard next-word prediction for smartphones indicated that a single user can benefit from personalization. P-meta (Qu et al., 2022) proposed a memory and computation efficient method for model adaptation by identifying a certain fraction of parameters that ensure rapid generalization and dynamically updating parameters to facilitate unseen tasks. Zhang (Zhang et al., 2021) relaxed the constraint of model homogeneity and aggregated the soft prediction of heterogeneous local devices using a knowledge group coefficient matrix to keep the global model updated with personalized context. However, these works did not extend to language tasks, which heavily rely on pretrained representation from the backbone model, and all assume that the personalized model is static in terms of model architecture and globally learned knowledge. In our work, we aim to investigate the preservation of personal information with model replacement

while minimizing the fine-tuning cost.

**Personalized Neural News Recommendation** News Recommendation is essential to mitigate information overload for users (Wu et al., 2022). It is widely employed in large news platforms like Google and Microsoft. The system generally consists of user modeling and news modeling. Using learned representations of users and news, the system predicts whether the news candidate will be clicked or not. In most works, the click decision is modeled by a similarity score between the encoded user's news history and the encoded news representation (Wu et al., 2021). Welch (Welch et al., 2022) used similar past user behaviors to learn a dynamic representation for new users. Interactive matching system (Qi et al., 2021) encoded user behaviors in the similarity of semantics and knowledge entities using a knowledge graph. We address the more challenging and dynamic scenario where the backbone is replaced and introduce user-level non-independent and identically distributed data into our work.

**Personalized Hate Speech Detection** Hate Speech Detection is a highly subjective tasks which leverages user-annotated data to perform supervised learning. Each person can have different considerations when labeling if a document is hateful (Paun et al., 2018; Röttger et al., 2022). Recent works identified the inherent diversity of hate speech data (Kanclerz et al., 2022) and empirically validated the user-specific perception based on personalized embedding and tailored classifiers. Since we focus on language-based classification tasks that inherently have user-specific distributions, we take personalized hate speech classification as our target task.

**Efficient Parameter Tuning** In the context of large language models, the training cost of fully tuning a model to a new domain from scratch is substantial. To preserve old knowledge and reduce fine-tuning efforts, methods, such as Adapter (He et al., 2021), Prefix-Tuning (Li and Liang, 2021) and LoRA (Hu et al., 2021), need to train only a small fraction of model parameters. These methods demonstrate improved performance in model fine-tuning compared to simply fine-tuning the top layer for downstream tasks, but they also encounter certain drawbacks. They often suffer from instability since their parameters need to be randomly initialized and require more training epochs to converge (Chen et al., 2022). In addition, the gradient computations of these methods during backpropagation are conditioned on the depth of the transformer blocks with adapters, yielding more floating point operations per second (FLOPS ) when training on hardware and also requiring memory to store the gradient. To address these drawbacks, our method only adds trainable parameters on the top layer and reduces gradient flow within the last prediction layer, which can significantly reduce the training cost in personalized fine-tuning.

## 3  Methodology

An edge device model provider regularly devises and trains new models on public or company-owned data. When deploying new models to edge devices, the goal is to minimize local tuning costs to achieve superior performance than the previous ones, while preserving privacy. In the following, the models trained on the public data is referred to as global models, while the models deployed to local devices are called local models.

### 3.1  Tasks

We conduct our experiments on two highly personalized tasks, News Recommendation (NR) and Hate Speech Detection (HS).

**News Recommendation** Our general framework is inspired by similarity matching (Wu et al., 2021) which utilizes the language model as shared backbone to encode news candidate and user history representations. For each user $u_i$, history behaviors are a set of news that they read denoted as $L = [l_1, l_2, ...l_n]$. A candidate news set, prepared to make predictions about user preference, is denoted as $R = [r_1, ..., r_m]$, where $m$ is the number of news candidates spanned by the user. A BERT based news encoder, along with an attention-based aggregation method, is then employed to generate representations for the news. News candidate representation is encoded as $v_c = NewsEncoder(l_c)$ and user representation as $u = UserEncoder(NewsEncoder(L))$. In first training phrase, the model is fine-tuned with globally collected user information. Given a user behavior list and a candidate news, we compute the inner product of two representations, $\hat{y} = u^T v_c$, as the prediction probability. We further incorporate negative sampling into the model training to calculate the posterior click probability for positive samples (Okura et al., 2017).

**Hate Speech Detection** We consider the classification of Hate Speech (HS) detection, where we use aggregated comments denoted as $R = [r_1, r_2..., r_m]$. Each is a text sequence for a document. The review representation is calculated using $v_c = NewsEncoder(l_c)$ (Kanclerz et al., 2022), and this representation is passed through a classification layer to calculate the prediction score $\hat{y} = w^T v_c$.

## 3.2 Task Formulation

Formally, given a text $\mathbf{x}_i$ with $\mathbf{x}_i \in \mathcal{X}$, we aim to build a model to predict the classification label $y_i \in \mathcal{Y}$, where $\mathcal{Y}$ denotes the label space. A model is a composite function $m(\mathbf{x}) = f \circ b_{t_0}(\mathbf{x})$, where $b_{t_0}(\mathbf{x})$ is referred to as the backbone model and $f(\mathbf{z})$ estimates labels of $\mathbf{x}$ based on the outputs of $b_{t_0}(\cdot)$. At time $t_0$, a global model $g_{t_0}(\mathbf{x}) = f_g \circ b_{t_0}(\mathbf{x})$ is trained on a public labeled dataset $\mathcal{D}_g = \{(\mathbf{x}_i, y_i)\}_{i=1}^N$ and its distribution is denoted by $p_g(Y|X)$. The globally trained model is deployed on a set of local devices. On each of such local devices $k$, there is a local dataset $\mathcal{D}_k = \{(\mathbf{x}_j, y_j)\}_{j=1}^{N_k}$ and $p_k(Y|X)$ characterizes the corresponding local distribution. On each device $k$, a local model $m_k(\mathbf{x}) = f_k \circ b_{t_0}(\mathbf{x})$ is created by fine tuning the global model on the local dataset without tuning the parameters of $b_{t_0}(\cdot)$. Furthermore, the distributions are different from each other such that $p_k(Y|X) \neq p_j(Y|X)$ if $k \neq j$, including the distribution of the public dataset. As a result, the optimal function $f_j(\cdot) \neq f_k(\cdot)$ if $j \neq k$. At time $t_i$, there is a significant update of the global model due to changes of the public dataset or introduction of new techniques. Therefore, the updated global model can use a totally different architecture than previous models, as long as the outputs of its backbone model are vector sequences. When deploying the new global model $g_{t_i}(\mathbf{x}) = f_{g,t_i} \circ b_{t_i}(\mathbf{x})$ to local devices, we aim to minimize the use of computing resources and obtain a new model $m_{k,t_i}(\mathbf{x}) = f_{k,t_i} \circ b_{t_i}(\mathbf{x})$ with optimal performance on each local device. Herein, the parameters of the backbone model $b_{t_i}(\cdot)$ are frozen during deployment. Due to privacy concerns, the local data on each local device is not used to train global models collaboratively.

## 3.3 Personalized Model Tuning and Matrix Replacement

At time $t_i$, when a local device receives a new global model, it replaces its current backbone with the new one $b_{t_i}(\cdot)$ and adaptively learns a new $f_{k,t_i}(\cdot)$ by using the logits derived from the previous model. The backbone model is usually referred to as the text encoder. The function $f_{k,t_i}(\cdot)$ essentially maps encoder outputs $\mathbf{h}$ to task-specific labels. For news recommendation, $f_{k,t_i}(\mathbf{h}) = \arg\max_y \log p(Y = y|\mathbf{h})$ with $y \in \{0, 1\}$. Herein, $\mathbf{h} = (\mathbf{u}, \mathbf{v})$, where $\mathbf{u}$ is the user representation and $\mathbf{v}$ denotes the representation of the current news. For a given news $c$, $p(Y = y|\mathbf{h}) = \sigma(z_c)$ and $z_c = \mathbf{u}^T \Lambda_k \mathbf{v}_c$, where the diagonal matrix $\Lambda_k$ captures user preference from the local distribution. Then $z_c$ is referred to as the logit of the news $c$. The rationale behind this is that the model intends to recommend the news similar to what a user read in the past. Hate speech detection is also a binary classification problem. The logit $z_c = \mathbf{w}_{t_i}^T \Lambda_k \mathbf{h}_c$, where $\mathbf{h}_c$ denotes the output from the backbone text encoder for a text $c$ and $\mathbf{w}$ is a weight vector shared between users to characterize the importance of each feature. The diagonal matrix $\Lambda_k$ encodes user-specific hate speech patterns.

Instead of using all instances from a local dataset, we build a fine-tuning dataset $D_k^f$ by sampling the most representative examples from a local distribution by applying correlation clustering. The details of correlation clustering are described in the following subsection.

During deployment, our key hypothesis is that the logits are similar before and after replacing the backbone model. Therefore, we save the logits $\hat{z}_i$ of the model before replacement for all instances in the fine-tuning dataset $D_k^f$, in order to compare them with the logits from the new model. As $\Lambda_k = \text{diag}(\theta)$ is a diagonal matrix for both applications, we can reformulate the learning problem after replacement as a linear regression problem by setting $z_i = \theta^T \mathbf{e}_i$ for all instances in $\mathcal{D}_k^f$, where $\mathbf{e}_i = \mathbf{u}_{i,t_i} \odot \mathbf{v}_{i,t_i}$ for news recommendation and $\mathbf{e}_i = \mathbf{w}_{i,t_i} \odot \mathbf{h}_{i,t_i}$ for hate speech detection. Because the backbone model is frozen, we need to make only one forward propagation for each instance in $\mathcal{D}_k^f$ to obtain the corresponding hidden representations. As a result, the adaptation process after replacement is a ridge regression problem below.

$$J(\theta) = \frac{1}{|\mathcal{D}_k^f|} \sum_i (z_i - \hat{z}_i)^2 + \lambda\|\theta\|_2^2 \quad (1)$$

The learned diagonal matrix should capture user-

specific information for respective applications.

**Correlation Clustering** To further reduce the number of document embeddings required to compute the personal matrix, we employ correlation clustering to cluster the documents read by the user. In our assumption, each user maintains their own list of documents and reading preferences, which means the clusters gathered for each individual can vary. By using correlation clustering, which does not need to specify the number of clusters, we obtain a varying number of clusters for each user. To cope with limited computing resources for a large document list, we choose the Kwikbucks method (Silwal et al., 2023), a query-efficient clustering method, as our clustering algorithm. It first picks pivots from a uniformly sampled subset of vertices from the graph and then gradually merges the neighboring nodes based on strong and weak signal pairwise relationships. In our constructed graph, we follow two rules to label the positive edge. Firstly, if the document pair have the same label for the user, for example, the user clicked on both of the documents or clicked on neither of them, we add a positive edge to this document pair, as shown in 2, where $i$ and $j$ are the indices of documents inside the user's document set.

$$E(d_i, d_j) = \begin{cases} 1, & \text{if } Y_{d_i} == Y_{d_j} \\ -1, & \text{otherwise} \end{cases} \quad (2)$$

Intuitively, a user who clicks on certain news may be more likely to click on documents with similar content. We also compute pairwise similarity of all documents that the user has read and mark the top $k$ similar documents for a certain document with a positive edge, as shown in 3.

$$E(d_i, d_j) = \begin{cases} 1, & \text{if } Cos(d_i, d_j) \in TopK(d_i, D) \\ -1, & \text{otherwise} \end{cases} \quad (3)$$

Together with the above two conditions, we construct a strong signal matrix with hard labels and a cosine similarity label with all pairwise similarities of the document set for each user, respectively. Thus, each user holds a cluster set that specifically belongs to them, denoted as $C_{U_1}$. We then choose examples from each cluster for linear optimization.

# 4 Experiment

## 4.1 Datasets

Our experiments utilizes two distinct datasets to perform the tasks of News Recommendation and Hate Speech Detection .

For News Recommendation , we conduct experiments using the Mind dataset (Wu et al., 2020) from Microsoft. This dataset consists of news articles and a corresponding log that tracks user behavior over a six-week period. Each log entry includes the click history of individual users, timestamped to provide temporal context. To facilitate personalization, we focus on the most active users in the dataset, selecting five for our study. These selected users have between 60 and 107 records each, indicating substantial interaction with the platform. We split the data for these users into training and testing subsets, with a 70:30 ratio. We use the small-size training dataset as our global training set. For Hate Speech Detection , we leverage the Wikipedia Detox Aggression dataset (Wulczyn et al., 2017), which contains 116k texts labeled by more than 4k annotators. The aggression score in the original dataset ranges from -3 to 3. As our focus is on the dynamic personal distribution of the tasks, we follow the setting from (Kanclerz et al., 2022) that emphasizes the significance of performing penalization. We convert the aggression scores to a binary system: 0 for non-aggressive content and 1 for aggressive content. Similar to our approach in the News Recommendation task, we identify the top five most active annotators, each with review counts ranging from 1382 to 1475, to create a personalized user set. All examples from picked users are not included in the global training datasets.

## 4.2 Experiment Settings

To simulate the training with updates between the cloud and local models, we divide the training process into three parts: Global Training, Personal Training, and Backbone Replacement Training.

**Global Training Phase:** In this phase, we use GloVe with 300 dimensions, BERT base size with 768 dimensions, and RoBERTa base model with 768 dimensions to initialize our backbones respectively. For each backbone model, we train models with globally collected examples and perform evaluations on personalized data. This establishes the baseline for individual users with different backbones. During training, we set the learning rate to

1e-5 and randomly pick 4 news articles as negative samples for News Recommendation . For hate speech, we use the same settings as above.

**Personal Training Phase:** In the personal training phase, we further train the personal model for individual users and test it with the corresponding test set. We use a personalized GloVe backbone as our old fine-tuned backbone, which is ready for backbone replacement. Additionally, we conduct full parameter model tuning for BERT and RoBERTa as a baseline, and perform FaLA for each user. To emphasize the significance of personalization, we repeatedly test all users across all personalized models to observe performance changes. As we are only concerned about the performance of penalization with replacement, we do not split personal data over more time intervals, which could be studied further.

**Backbone Replacement Training:** For the personalized model, we replace the outdated backbone with a new one. We switch the personalized backbone to a globally trained new backbone, specifically, from GloVe to BERT and GloVe to RoBERTa . We insert the personalized computed diagonal matrix into the top layer of the new global model for the corresponding users. Then, we carry out the same procedure as in personal training but only tune the parameters within the personal matrix.

## 4.3 Baseline

We consider five baselines to perform personalized model tuning for each user, as depicted in Table 2.

**Global Model** (Global): It directly applies the new global backbone model to local devices.

**Full parameter tuning** (Full): It fine-tunes all parameters of the new global model locally.

**Top layer fine tuning** (Top): This method fine-tunes only the parameters of the task-specific layers, which is widely used for most on-device models (Xu et al., 2018).

**Adapter** : Adapter (He et al., 2021) used inside the BERT model is popular in transfer learning for language models to reduce the training parameters inside the backbone.

**LoRA** : LoRA (Hu et al., 2021) constructs low rank decomposition for BERT 's high dimensional weights inside the language model, claiming the state-of-the-art performance for model fine tuning.

## 4.4 Evaluation Metrics

### 4.4.1 Model Effectiveness

We compare the performance of state-of-the-art News Recommendation systems (Wu et al., 2021, 2019) with our proposed method on personalized users. Following previous work on News Recommendation metrics (An et al., 2019; Wu et al., 2019), we use AUC as an evaluation metric. For Hate Speech Detection , we use accuracy and F-1 score (Kanclerz et al., 2022) as our metrics to illustrate the effectiveness of the personalized classification model.

### 4.4.2 Model Efficiency

To evaluate computational efficiency, we record the FLOPS (Floating Point Operations Per Second) for each personal training setting. FLOPS is a measure of scientific computation capability of a device. Language models containing a large number of parameters use FLOPS as an indicator of their computation cost. A model that achieves similar performance with fewer FLOPS is generally considered more efficient. We employ the Tensor-Flow profiler (Abadi et al., 2015) to record forward and backward computation of a training batch on an NVIDIA 3090 to illustrate the computational differences among the tested methods.

## 4.5 Implementation Details

Overall, our experiments are carried out on an NVIDIA 3090 24G GPU over multiple rounds. We use hyperparameters suggested from previous work to establish the baseline. In addition, we mainly tuned three hyperparameters during the training process: model learning rate $\eta$, $\lambda$ for L2 regularization, top $k$ similarity examples used to construct a strong signal relation group, and regularization selection for linear optimization. As these three parameters have a significant impact on the downstream task, we use a grid search to obtain the optimal values for these terms while keeping the other hyperparameters constant. For News Recommendation , we use the top 10 similar examples, a 1e-3 learning rate for linear optimization and 1e-5 for backbone tuning, and set $\lambda$ to 1e-1. For Hate Speech Detection , we use 1e-3 for linear optimization, 1e-4 for backbone tuning, and set the L2 regularization term to 1e-4. For the impact of different selections, we further discuss this in 5.

Table 1: Measured FLOPS of one training batch for News Recommendation

| Adapter Method | BERT | | RoBERTa | |
|---|---|---|---|---|
| | Forward | Backward | Forward | Backward |
| base | 1.3563e+12 | 2.7100e+12 | 2.7127e+12 | 5.4200e+12 |
| Adapter | 1.3941e+12 | 1.4000e+12 | 2.7882e+12 | 2.8000e+12 |
| LoRA | 1.3613e+12 | 1.3500e+12 | 2.7226e+12 | 2.6900e+12 |
| Ours | 1.3563e+12 | **5.4500e+09** | 2.7127e+12 | **1.0900e+10** |

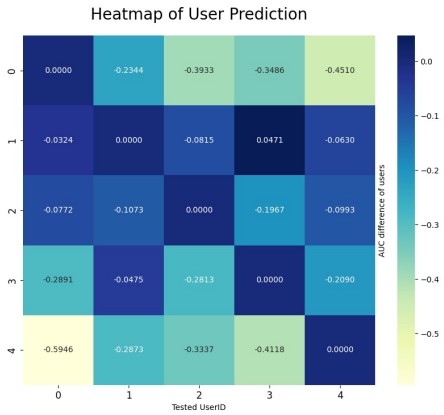

Figure 2: Performance difference for individuals

## 4.6 Experiment Results

### 4.6.1 Measured FLOPS

As presented in Table 1, the computational efficiency of the forward pass per batch does not show considerable escalation, owing to the relative insignificance of the added parameters compared to the full parameter size tuning of the base models. For backpropagation, full parameter tuning yields almost 2 times the FLOPS compared to forward computation. Considering the Adapter and LoRA methods, the computational cost nearly mirrors that of the forward pass. This can be attributed to the backpropagation of the gradient into the transformer blocks equipped with adapter modules. Since our method is only applied to the top layer and involves only one layer gradient computation, this leads to a striking improvement in computational efficiency during backpropagation training, making our approach over 100 times more efficient than both the Adapter and LoRA methods. This underlines the superiority of our method in terms of computational efficiency.

### 4.6.2 General Performance

**Personal Training** The experiments comparing globally trained models to personalized models demonstrate the effectiveness of employ-

ing personalized training. For the 5 most frequent users from News Recommendation and Hate Speech Detection , we recorded their average performance and standard deviation in terms of AUC for News Recommendation , Accuracy and F-1 for Hate Speech Detection . We firstly observed the impact of personalization in our old backbone GloVe as shown in table 3, the personalization improves the performance of GloVe based models when encountering new users, indicating the personal preference learned by the system. As shown in Table 2, fully parameter-fine-tuned models consistently outperform the globally trained models. Moreover, for each personalized model, we tested it across the data of all users to show the performance differences for the tuned model using a heat map as shown in Figure 2. Each row in the heat maps represents the performance difference of all users tested by one users' personalized model. The distinguished prediction for each user further illustrates the inherent distribution difference, enhancing the necessity of performing personalization.

**Personalized Tuning** For personalized fine tuning, as Table 2 depicts, if we only tune the task specific layers without additional parameters added in, the user domain adaption ability is poor for all metrics. Adapter failed to adapt to user domain with limited examples. From the table 2, it is evident that our proposed method outperforms the other techniques in terms of AUC and F-1 score. Notably, the performance improvement is not marginal but significant, reinforcing the efficacy of our approach. Specifically, compared to the Adapter , our method shows an improvement of approximately 6.46%,7.81%, and against the LoRA , the enhancement is about 2.8%,2.26% for BERT andRoBERTa in News Recommendation respectively. In Hate Speech Detection , FaLA consistently outperforms all baselines at least over 6% in F-1 for both backbones. These significant improvements underscore the superiority of our approach.

## 5 Ablation Study

To further explore and validate our methods, we conduct ablation study focusing on diverse factors throughout our personalized matrix computation process.

Table 2: Overall Performance

| Tasks | Model | Metrics | Global | Full | Adapter | LoRA | Top | FaLA |
|-------|-------|---------|--------|------|---------|------|-----|------|
| NR | BERT | AUC | $57.45 \pm 3.05$ | $63.07 \pm 10.25$ | $61.01 \pm 13.11$ | $64.67 \pm 11.27$ | $61.39 \pm 12.83$ | $\mathbf{67.47} \pm 13.65$ |
| NR | RoBERTa | AUC | $60.11 \pm 1.27$ | $60.50 \pm 11.99$ | $57.81 \pm 9.25$ | $60.98 \pm 17.28$ | $63.38 \pm 12.50$ | $\mathbf{65.64} \pm 5.26$ |
| HS | BERT | ACC | $43.97 \pm 23.14$ | $87.02 \pm 3.62$ | $66.57 \pm 16.12$ | $86.61 \pm 5.03$ | $64.85 \pm 15.15$ | $85.98 \pm 5.67$ |
| HS | BERT | F1 | $25.70 \pm 8.82$ | $44.00 \pm 11.69$ | $18.20 \pm 8.82$ | $41.55 \pm 18.46$ | $22.94 \pm 10.98$ | $\mathbf{50.94} \pm 6.49$ |
| HS | RoBERTa | ACC | $24.84 \pm 27.26$ | $87.74 \pm 4.81$ | $57.88 \pm 31.48$ | $87.54 \pm 4.30$ | $54.06 \pm 33.30$ | $86.17 \pm 6.83$ |
| HS | RoBERTa | F1 | $17.97 \pm 10.21$ | $40.83 \pm 21.21$ | $16.88 \pm 15.77$ | $39.21 \pm 23.51$ | $10.40 \pm 9.10$ | $\mathbf{48.42} \pm 11.23$ |

Table 3: Performance of Global and Personalized GloVe

| Task | Metric | Global | Personal |
|------|--------|--------|----------|
| NR | AUC | $45.42 \pm 8.11$ | $53.14 \pm 7.29$ |
| HS | ACC | $49.41 \pm 33.52$ | $83.86 \pm 6.90$ |
| HS | F-1 | $32.11 \pm 22.46$ | $37.52 \pm 20.21$ |

## 5.1 Validating Personalized Matrix Tuning

In our design, we added personalized matrix when updating the model, we would like to answer two main questions: Is the initialization of a personalized matrix essential? If so, how does the quality of our clustering compare to conventional cluster method? To derive results, we considered the following scenarios:

- **Random Initialization** The matrix parameters are initialized randomly to examine whether the matrix inserted into the top layer is effective.

- **Full Rank Example Optimization** The matrix is trained in full rank with a multitude of examples gathered from the users' log, equivalent to the dimension of the matrix.

- **K-Means** We implement k-means in our example selection process as it is a fast and popular clustering method.

- **Our method with K examples** To justify the performance with K-means, we compiled samples from the top k clusters using FaLA, and tuned the model with the same quantity of examples as K-means.

- **FaLA** The baseline performance for FaLA.

As per the results shown in Table 4, we observed that directly applying the matrix to the top layer does not sufficiently impact training in most cases. The method's performance is the least effective as it

increases the training loss and demands extra training effort for both tasks. Our empirical findings show that the optimal K for K-means for BERT based both tasks is 50 and 10 for RoBERTa based tasks. Surprisingly, for BERT based models, FaLA with K examples showcased the best performance, surpassing the FaLA for News Recommendation. With the collected cluster number from FaLA being less than 50, we randomly sampled the $50 - K$ examples from the remaining examples to construct the training examples for linear optimization. This resulted in a superior personal initialization. For the RoBERTa based model, which limited 10 examples for training, our method exhibited competitive results for our method in 10 examples compared to K-means. And FaLA demonstrated comparable performance in all cases with respect to the full rank matrix initialization.

## 5.2 Evaluation of Adaptation Objectives

**Choices of Linear Objectives** We explored the influence of diverse linear objective functions on the system's performance. Each combination of objectives denotes a unique learning approach, which can significantly impact the results. The prediction of our system is logits that can be further converted to probabilities using the sigmoid function. As indicated in Table 5, we individually eliminated each objective function to observe the subsequent performance changes. The objectives evaluated included Mean Square Error(MSE) and Cross Entropy(CE) both with and without L2 regularization. We found that combining these two terms led to no noticeable performance difference, suggesting that this objective combination may not be crucial to our model's performance. However, when we solely applied MSE to optimization, it resulted in the best performance for both tasks, underscoring the effectiveness of logits mapping for clustered examples.

Table 4: Matrix Initialization Methods

| Task | Model | Metric | Random Initial | Full Rank | K-Means | FaLA-K | FaLA |
|------|-------|--------|----------------|-----------|---------|--------|------|
| NR | BERT | AUC | 57.45% | 67.48% | 66.29% | **69.16**% | 67.47% |
| NR | RoBERTa | AUC | 60.11% | 62.76% | 59.66% | 61.94% | **65.64** % |
| HS | BERT | ACC | 82.71% | 82.26% | 84.35% | 85.95% | **85.98**% |
| HS | BERT | F1 | 42.43 % | 48.11% | 41.30% | 42.93% | **50.18** % |
| HS | RoBERTa | ACC | 84.24% | 79.14% | 84.86% | 85.94% | **86.17** % |
| HS | RoBERTa | F1 | 46.25% | 51.55% | 43.42% | 46.85% | 48.42% |

Table 5: Selection of Objective Term

| TASK | Model | Lambda | CE L2 | MSE L2 | MSE L2 CE |
|------|-------|--------|-------|--------|-----------|
| NR | BERT | 0.0001 | 62.37% | 67.47% | 65.16% |
| NR | BERT | 0.001 | 67.74% | 65.34% | 63.31% |
| NR | BERT | 0.01 | 67.26% | 62.91% | 64.35% |
| NR | BERT | 0.1 | 67.31% | **69.05**% | 65.27% |
| NR | BERT | 0 | 68.28% | 68.65% | 64.61% |
| HS | BERT | 0.0001 | 84.14% | **86.33**% | 83.58% |
| HS | BERT | 0.001 | 84.57% | 83.54% | 84.08% |
| HS | BERT | 0.01 | 85.59% | 81.62% | 81.34% |
| HS | BERT | 0.1 | 82.09% | 85.41% | 82.84% |
| HS | BERT | 0 | 84.60% | 84.64% | 84.90% |

**Hyperparamemter Sensitivity** We also engaged in a sensitivity analysis of the system's performance in relation to hyperparameters, particularly focusing on the L2 regularization term. This term is pivotal in controlling the model's complexity and preventing overfitting. Given that our method necessitates only a handful of examples to optimize high-dimensional parameters, we intended to calibrate the weight decay factor to avert overfitting during the learning process. We observed a noticeable trend whereby an incremental increase in the L2 term from lower to higher values corresponded with an enhancement in the model's accuracy for Cross Entropy (CE) when evaluated individually. Meanwhile, the Mean Square Error (MSE) displayed optimal performance when the regularization strength is set to 0.1 for News Recommendation and 0.0001 for Hate Speech Detection . This underscores the significance of carefully selecting this hyperparameter to ensure both stability and accuracy in our model's predictions.

## 6 Conclusion

In this paper, we introduce FaLA , the fast linear adaptation method for backbone model replacement. We consider a novel and realistic dynamic training scenario that the service providers want to minimize the computational cost of updating the backbone of the on-device models deployed to their client devices while preserving privacy of the local data. Specifically, we reformulate the complex model replacement problem as a simple Ridge regression problem. Our approach notably outperforms established baseline techniques in terms of both accuracy and computational efficiency. This work identifies new research problems regarding backbone replacement. For future work, we may move beyond classification tasks to broader applications with diverse backbones.

## Limitations

The limitations of our work are three-fold:

Firstly, our experiments are primarily conducted on a subset of datasets. The performance, efficiency, and efficacy of parameter tuning may vary across different types of datasets. In our experiments, replaced backbone models are BERT and RoBERTa . More comprehensive experiments with more backbones may make the empirical results more solid.

Secondly, hyperparameter search is still time-consuming for each user. More efficient methods may be applied to reduce computational cost.

Thirdly, in our experiments, we apply one forward pass to each user example and store the corresponding representations in disk to build clusters. This is regarded as a way of trading off the storage cost with the computational time. Further reduction of the storage cost remains an open question.

To sum up, our work investigates a practical scenario regarding efficient model deployment for edge devices. The aforementioned limitations still pose potential areas for future research.

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

# A  Appendix

## A.1  User Clsuter States

We record the number of optimal k similarity for all models and tasks shown in table 7. From the table, we can see that for News Recommendation task with small value of topk to construct the graph, there is fewer postitive edge constcuted therefore resulting more clusters. For Hate Speech Detection , the optimal value is relatively high, the number of cluster center is less.

## A.2  Heat Map for Hate Speech Detection

We present the heat map of hate speech detection to show the performance of trained user-specific classifiers. As depicted in Figure 3, each classifier act in a distinguished manner demonstrating the existence of user specific distributions.

## A.3  Training without Global Data

We also explored the setting with directly fine tuning the model with user's data respectively. As illustrated in Table 6, solely relying on fine-tuning without global training led to models that were inadequately personalized and exhibited considerable performance variation. For instance, while our method achieved an AUC of 65.64% for news recommendation, the result for a model that was only fine-tuned stood at 52.44%. This significant performance discrepancy was consistent across various settings.

The rationale behind this discrepancy is that task-specific global training embeds a broader knowledge base regarding the tasks into the model. Subsequent personalized training can then adjust this generalized knowledge to cater to individual user

Table 6: Fine Tuned Model with GloVe

| Task | Metric | BERT | RoBERTa | BERT -FaLA | RoBERTa -FaLA |
|------|--------|------|---------|------------|---------------|
| NR | AUC | 55.86% ± 4.05% | 52.44% ± 12.48% | 67.47%±13.65% | 65.64%±5.26% |
| HS | F-1 | 40.93% ± 12.41% | 42.51%± 24.76% | 50.94%±6.49% | 48.42%± 11.23% |

Table 7: User Clusters

| TASK | MODEL | Topk Similarity | User 1 | User 2 | User 3 | User 4 | User 5 |
|------|-------|-----------------|--------|--------|--------|--------|--------|
| NR | BERT | 10 | 42 | 100 | 112 | 429 | 90 |
| NR | RoBERTa | 10 | 42 | 100 | 68 | 418 | 104 |
| HS | BERT | 250 | 15 | 8 | 24 | 26 | 8 |
| HS | RoBERTa | 250 | 21 | 12 | 19 | 20 | 20 |

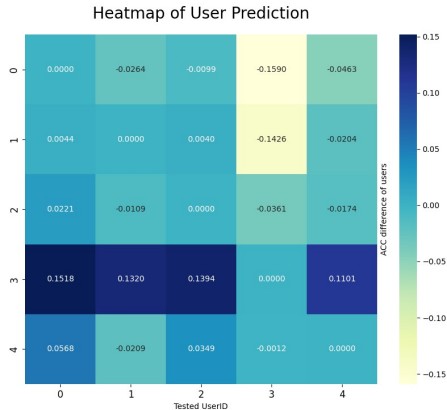

Figure 3: Performance difference for individuals in Hate Speech Detection

distributions. However, in the absence of a substantial dataset for a specific user, accurately modeling such a user-specific distribution becomes immensely challenging. This is compounded by the fact that local devices have constraints in terms of data storage capacity and require periodic updates.