# OpenReview forum: "FaLA: Fast Linear Adaptation for Replacing Backbone Models on Edge Devices"
_EMNLP/2023/Conference — EMNLP 2023 Findings_

### Official Review · Reviewer_y97i · 2023-08-02

**Typos Grammar Style And Presentation Improvements:** 004
**Soundness:** 3

**Excitement:**

2: Mediocre: This paper makes marginal contributions (vs non-contemporaneous work), so I would rather not see it in the conference.

**Missing References:**

refer to reasons to reject

**Paper Topic And Main Contributions:**

The paper studies the problem of language model backbone replacement for personalized downstream tasks in a non-stationary on-device scenario and proposes a lightweight tuning method to solve it. The proposed method achieves over 1000x computation reduction in Flops for back-propagation and brings significant performance boost compared with popular transfer learning methods.

**Questions For The Authors:**

refer to reasons to reject

**Reasons To Accept:**

1. The topic of this paper studied is important and interesting.
2. The proposed method is lightweight and can be easily implemented on-device, which is important for real-world applications.
3. The method achieves significant performance improvements over popular transfer learning methods.

**Reasons To Reject:**

1. The writing and presentation quality are not good enough to be accepted in EMNLP. First, the problem definition is not clear. Please strictly define the problem to be solved in mathematical terms, including the data in global and local respectively and the goal of the paper. Please summarize the whole training flow. Please add more description about Figure 1. It is very difficult for readers to guess the meaning of the components in the figure.
2. The paper proposes methods for on-device applications. Therefore, it is necessary to report the memory used for each method. And for back-propagation computation reduction, it is better to show the actual running time of each method. Just reporting FLOPs is not convincing enough. Besides, is the clustering step included in FLOPs computation?
3. The paper lacks related work about federated learning. The parameter-efficient personalized federated learning is very similar to the setting in this paper. However, the paper does not show related work about parameter-efficient personalized federated learning. And the paper should add one baseline [1] or [2].

[1] Pillutla, Krishna, et al. "Federated learning with partial model personalization." International Conference on Machine Learning. PMLR, 2022.
[2] Xie, Chulin, et al. "PerAda: Parameter-Efficient and Generalizable Federated Learning Personalization with Guarantees." arXiv preprint arXiv:2302.06637 (2023).

**Reproducibility:**

4: Could mostly reproduce the results, but there may be some variation because of sample variance or minor variations in their interpretation of the protocol or method.

**Reviewer Confidence:**

4: Quite sure. I tried to check the important points carefully. It's unlikely, though conceivable, that I missed something that should affect my ratings.

---

> ### Author Rebuttal · Authors · 2023-08-29
>
> Dear Reviewer,
>
> Thank you for taking the time to provide a thorough review of our submission. We value your feedback and have taken each comment seriously. We are making significant revisions to address the concerns raised.
>
> 1. Writing and presentation quality, diagram quality:
>
> Response: We acknowledge the feedback regarding the clarity and presentation of our work. To address this:
>
> We've revisited the problem definition and provided a rigorous mathematical formulation as the following.
>
> we aim to build a model to predict the classification label $y_i \in \mathcal{Y}$, where $\mathcal{Y}$ denotes the label space. A model is a composite function $m(\mathbf{x}) = f \circ b_{t_0}(\mathbf{x})$, where $b_{t_0}(\mathbf{x})$ is referred to as the backbone model and $f(\mathbf{z})$ estimates labels of $\mathbf{x}$ based on the outputs of $b_{t_0}(\cdot)$. At time $t_0$, a global model $g_{t_0}(\mathbf{x}) = f_g \circ b_{t_0}(\mathbf{x})$ is trained on a public labeled dataset $\mathcal{D}_g = \{(\mathbf{x}_i, y_i)\}_i^N, i=1$ and its distribution is denoted by $p_g(Y|X)$.  The globally trained model is deployed on a set of local devices. On each of such local devices $k$, there is a local dataset $\mathcal{D}_k = \{(\mathbf{x}_j, y_j)\}_j^{N_k}, j=1$ and $p_k(Y|X)$ characterizes the corresponding local distribution.  On each device $k$, a local model $m(\mathbf{x}) = f \circ {b_t}_0(\mathbf{x})$, is created by fine tuning the global model on the local dataset without tuning the parameters of ${b_t}_0(\cdot)$. Furthermore, the distributions are different from each other such that $p_k(Y|X) \neq p_j(Y|X)$ if $k \neq j$, including the distribution of the public dataset.   As a result, the optimal function $f_j(\cdot) \neq f_k(\cdot)$ if $j \neq k$. At time $t_i$, there is a significant update of the global model due to changes of the public dataset or introduction of new techniques. Therefore, the updated global model can use a totally different architecture than previous models, as long as the outputs of its backbone model are vector sequences. When deploying the new global model to local devices, we aim to minimize the use of computing resources and obtain a new modelwith optimal performance on each local device. Herein, the parameters of the backbone model ${b_t}_i(\cdot)$ are frozen during deployment. Due to privacy concerns, the local data on each local device is not used to train global models collaboratively.
>
>
> At time $t_i$, when a local device receives a new global model, it replaces its current backbone with the new one $b_{t_i}(\cdot)$ and adaptively learns a new $f_{k,t_i}(\cdot)$ by using the logits derived from the previous model. The backbone model is usually referred to as the text encoder. The function $f_{k,t_i}(\cdot)$ essentially maps encoder outputs $\mathbf{h}$ to task-specific labels. For news recommendation, $f_{k,t_i}(\mathbf{h}) = \arg\max_y \log p(Y = y | \mathbf{h})$ with $y \in \{0, 1\}$. Herein,  $\mathbf{h} = (\mathbf{u}, \mathbf{v})$, where $\mathbf{u}$ is the user representation and $\mathbf{v}$ denotes the representation of the current news. For a given news $c$, $p(Y = y | \mathbf{h}) = \sigma (z_c)$ and $z_c = \mathbf{u}^{\text{T}} \Lambda_k \mathbf{v}_c$, where the diagonal matrix $\Lambda_k$ captures user preference from the local distribution. Then $z_c$ is referred to as the logit of the news $c$. The rationale behind this is that the model intends to recommend the news similar to what a user read in the past. Hate speech detection is also a binary classification problem. The logit $z_c = \mathbf{w_t}_i^{\text{T}} \Lambda_k \mathbf{h}_c$, where $\mathbf{h}_c$ denotes the output from the backbone text encoder for a text $c$ and $\mathbf{w}$ is a weight vector shared between users to characterize the importance of each feature. The diagonal matrix $\Lambda_k$ encodes user-specific hate speech patterns.
>
>
> Instead of using all instances from a local dataset, we build a fine-tuning dataset $D_k^f$ by sampling the most representative examples from a local distribution by applying correlation clustering. The details of correlation clustering are described in Line 263 - 303.
>
>
> During deployment, our key hypothesis is that the logits are similar before and after replacing the backbone model. Therefore, we save the logits $\hat{z}_i$ of the model before replacement for all instances in the fine-tuning dataset $D_k^f$, in order to compare them with the logits from the new model. As $\Lambda_k = \text{diag}(\theta)$ is a diagonal matrix for both applications, we can reformulate the learning problem after replacement as a linear regression problem by setting $z_i = \theta^{\text{T}}\mathbf{e}_i$ for all instances in $\mathcal{D}^f_k$. Because the backbone model is frozen, we need to make only one forward propagation for each instance in $\mathcal{D}^f_k$ to obtain the corresponding hidden representations.
> The learned diagonal matrix should capture user-specific information for respective applications. Because the backbone model is frozen, we need to make only one forward propagation for each instance in $\mathcal{D}^f_k$ to obtain the corresponding hidden representations. As a result, the adaptation process after replacement is a ridge regression problem below.
>
> $$
> J(\theta) = \frac{1}{|\mathcal{D}_k^f|} \sum_{i} (z_i - \hat{z}_i)^2 + \lambda \| \theta \|_2^2
> $$
>
> Thank you for your feedback regarding Figure 1. In response to your comments, we have updated the figure to offer a more transparent representation of the system workflow and the associated problem. Specifically:
>
> We've detailed the model replacement process by using a more concrete example and illustrating the transition from GloVe to BERT explicitly. The workflow now incorporates a visual representation of our method. We've added additional steps highlighting how correlation clustering is employed to gather data and how the optimization of the personal linear matrix takes place. Subsequent to these steps, the process of personal matrix fine-tuning has been visualized, demonstrating how it facilitates adaptation to the personalized new backbone model. We've enriched the figure with supplementary documentation and descriptions to ensure every phase is straightforward and intuitive for the readers.
>
>
> We hope these revisions address your concerns and make the figure more informative and clear.
>
>
> 2.memory report and flops for correlation clustering?
>
>
>
>
> Model Memory: We implement our method using TensorFlow, thus the GPU memory for models is pre-allocated before training and inference. The allocation of GPU memory was 2900 MB in total. %Discerning a direct difference in memory usage becomes challenging due to this preloading mechanism. However we can estimate the memory usage for each stage given the existing condition mathematically.
> Because we cannot direct observe actual GPU memory usage at each stage, we estimate it based on the expected consumption. For bert-base-model with 768 dimensions with token length of 768 with batch size of 150 examples
> We can calculate the activation memory for forward pass as $Forward Pass Memory=150×25×768×4×12 \approx 1.1G$ and for backward pass we need extra memory to store the gradient and weight. For full parameter tuning it would be $Backward Pass Memory (Full Model)≈Forward Pass Memory \approx 1.1G$. If we only tune last layer in gradient checkpoint it would be $Backward Pass Memory (Top Layer)=150×25×768×4
> \approx 92M$. There will be overall around 400M for static parameters of models.
>
> So, for both forward and full backward pass, you'd need roughly 2.2 GB + 440 MB = ~2.64 GB. For forward pass and top layer backprop, you'd need roughly 1.1 GB + 92 MB + 440 MB = ~1.63 GB. As this is case of training on service, with much less batch size of personal training on deivce for example 3 examples, the computation can be largely reduced and fit to on-device scinerio.
>
>
>
> Flops for the Correlation Clustering Method (Kwikbucks): Present techniques for monitoring Flops are predominantly oriented towards deep learning methods, making a direct measurement for our correlation clustering method impractical. Nevertheless, we can offer insights into the execution time of the correlation clustering algorithm. Each clustering iteration is completed in approximately 2 seconds. This duration is notably shorter than what's needed for either a forward pass or a backward pass.
> We trust that this elucidation aptly addresses your concerns and presents the information more succinctly.
>
>
>
> 3. Related work on federated learning:
>
> Response:
>
> For the version we submitted, we opted to omit this section due to space constraints and for the following reasons:
>
> Federated learning aims to collaboratively learn models in a distributed manner. However, our problem setting does not consider learning a model collaboratively by using the data from local devices. Instead, we consider only the deployment of a globally trained model to local devices without using any model updates or local data from them. Moreover, we consider the adaptation process by replacing one component of the local models, not all of them, while most federated learning applications consider shared model architectures between servers and local devices. Our setting is also different from the FL works that us smaller versions of models on local devices. We do not consider model pruning or model compression but only want to accelerate the deployment process using as few computing resources as possible.
>
> Personalized parameter-efficient tuning strategies, such as the adapter mechanism or partial parameter updates as mentioned by scholars[2,4,5], are not directly applicable to our method, because they assume the same or similar architecture for the backbone models. Their methods will fail if the dimension of the backbone model outputs changes vastly. For instance, transitioning from an old model with a dimension of 512 to a new one with 768. The dimension mismatch prevents from a direct integration of the old model's upper layers into the new backbone model.
>
>
>
> As in our first version of paper, we do include the personalized federated learning in our related work the original paper is attached here:
>
> "Wang[1] evaluated on-device personalization under a federated learning setting where personal data is preserved on local devices. Their experiment on virtual keyboard next-word prediction for smartphones indicated that a single user can benefit from personalization.
> P-meta[2] proposed a memory and computation efficient method for model adaptation by identifying a certain fraction of parameters that ensure rapid generalization and dynamically updating parameters to facilitate unseen tasks.
> Zhang[3] relaxed the constraint of model homogeneity and aggregated the soft prediction of heterogeneous local devices using a knowledge group coefficient matrix to keep the global model updated with personalized context.
> However, these works did not extend to language tasks, which heavily rely on pretrained representation from the backbone model, and all assume that the personalized model is static in terms of model architecture and globally learned knowledge. In our work, we aim to investigate the preservation of personal information with model replacement while minimizing the fine-tuning cost."
> However, these studies didn't venture into language-centric tasks that are heavily dependent on pretrained representations from primary models. They also operated under the assumption that the personalized model remains constant in terms of architecture and globally acquired knowledge."
>
>
> Considering the feedback, we're contemplating the addition of an in-depth section on parameter-efficient personalized federated learning, either in the related work or appendix. This will help to distinctly highlight the differences and underscore our research objective.
>
> We hope this revision provides clearer insights into our decision-making process and the distinctive aspects of our research.
> reference:
>
> [1]Wang, K., Mathews, R., Kiddon, C., Eichner, H., Beaufays, F., & Ramage, D. (2019). Federated evaluation of on-device personalization. arXiv preprint arXiv:1910.10252.
>
> [2]Qu, Z., Zhou, Z., Tong, Y., & Thiele, L. (2022). p-Meta: Towards On-device Deep Model Adaptation. In 28th ACM SIGKDD International Conference on Knowledge Discovery and Data Mining (KDD 2022) (pp. 1441-1451). Association for Computing Machinery.
>
> [3]Zhang, J., Guo, S., Ma, X., Wang, H., Xu, W., & Wu, F. (2021). Parameterized Knowledge Transfer for Personalized Federated Learning. Advances in Neural Information Processing Systems, 34, 10092-10104.
>
> [4] Pillutla, K., Malik, K., Mohamed, A. R., Rabbat, M., Sanjabi, M., & Xiao, L. (2022, June). Federated learning with partial model personalization. In International Conference on Machine Learning (pp. 17716-17758). PMLR.
>
> [5] Xie, C., Huang, D. A., Chu, W., Xu, D., Xiao, C., Li, B., & Anandkumar, A. (2023). PerAda: Parameter-Efficient and Generalizable Federated Learning Personalization with Guarantees. arXiv preprint arXiv:2302.06637.

---

### Official Review · Reviewer_GxQw · 2023-08-05

**Soundness:** 3

**Excitement:**

4: Strong: This paper deepens the understanding of some phenomenon or lowers the barriers to an existing research direction.

**Paper Topic And Main Contributions:**

This paper focuses on the problem of language model backbone replacement for personalized training in the cloud edge scenario. In real applications, user data is stored on the local device alongside the task-specific parameters trained on that data whereas the main base/backbone model is stored within the cloud. This is particularly important for the personalized use case due to privacy concerns, the efficiency of model training as well as the distribution discrepancies between user data and global data. Over time this base backbone model is updated, resulting in the need to retrain all local models across users. This paper proposes the task of efficiently personalizing the model after the backbone is replaced. They introduce Fast Linear Adaption (FaLA) where they preserve old logits and compute a new personalized matrix based on the hidden representations of the new model. They showcase how this method is computationally more efficient compared to existing parameter-efficient fine-tuning methods.

**Questions For The Authors:**

- Is the backbone of the model frozen when producing the first personalized local model at time $P_{t_0}$ ?
- Did you explore any experimentation without training the global model on all user's data? Is there a clear separation between the data using the global training phase and the personal training phase?
- Why do you only select the top 5 users? Does this mean the global model is trained on the data for just those 5 users?
- It's surprising that Full would perform worse. Why is this? When you say it finetunes all parameters of the new model, I'm assuming this is for each user correct?
- Line 497 - 500 You mentioned this shows applicability to new unseen users. In which experimentation is this explored?
- What is the performance gap across time gaps? How did FaLA perform at timestep 0 compared to timestep 1?

**Reasons To Accept:**

- The problem scenario is interesting and has real-world implications. Keeping a clear separation between cloud and edge is vital for the efficiency and privacy of ML models on low-powered devices.
- Initial experimentation results show that FaLA is more computationally efficient and can outperform comparable PEFT methods.

**Reasons To Reject:**

- Diagramming needs work, Figure 1 does a poor job of providing the user with any understanding of the system workflow or the problem at hand. While interesting overall, the paper does seem rushed.
- Experimentation is unclear: limited information is provided on the main focus of the paper which is "Backbone Replacement Training".
- Applicability of the method to generative tasks is not explored in experimentation. As such I would advise the authors to revise the wording to make it explicit that this approach is geared toward encoder-based classification tasks.

**Reproducibility:**

3: Could reproduce the results with some difficulty. The settings of parameters are underspecified or subjectively determined; the training/evaluation data are not widely available.

**Reviewer Confidence:**

4: Quite sure. I tried to check the important points carefully. It's unlikely, though conceivable, that I missed something that should affect my ratings.

---

> ### Author Rebuttal · Authors · 2023-08-29
>
> Dear Reviewer,
>
> Thank you for the detailed review and constructive feedback on our submission. We sincerely appreciate the time and effort you spent in reviewing our paper. We have addressed each of your concerns and made appropriate revisions to improve the clarity and quality of the paper. Below, we provide a point-by-point response to your comments.
>
> We would like to respond your review for reasons to reject and questions respectively.
> 1. Diagramming needs work:
>
>
> Response:
>
> We acknowledge your concern regarding Figure 1. Based on your feedback, we have revised the figure to provide a clearer representation of the system workflow and the problem at hand. We have the following adjustment to our digram:
>
> We have updated the figure to more clearly delineate the system's workflow and the central issue we're addressing.
> To elucidate the model replacement process, we've highlighted the shift from GloVe to BERT more distinctly within the revised figure.
> Our approach is now visually integrated into the workflow to offer a more comprehensive view.
> Additionally, we have incorporated steps that detail how correlation clustering is utilized for data collection and how the personal linear matrix is optimized.
> Subsequently, we illustrate the fine-tuning of the personal matrix to align with the updated, personalized backbone model.
> To ensure clarity, added annotations and explanations accompany each step, making the process more intuitive for readers.
>
> We trust that these revisions more accurately address your comments and enhance the figure's interpretability.
>
>
>
> 2. Experimentation is unclear:
>
> Response:
>
> We apologize for any confusion caused. we are describing the phase in which the model backbone is substituted, and the local device begins personalizing the newly integrated on-device model. Our training processes are in three distinct stages, capturing the transformations both preceding and succeeding the backbone replacement. The primary objective of this stage is to fine-tune the new model locally, leveraging private data.
>
> Central to our discourse is the acceleration of the fine-tuning process, harnessing pre-existing information to minimize associated costs. Our whole methodology is operational during the "backbone replacement training" phase. To elucidate, we have incorporated a new workflow. This schematic underscores the chronology and mechanisms of our process, highlighting the steps at which we undertake example clustering, linear optimization, and subsequent model training. We believe this visual aid enhances clarity and provides a more intuitive understanding of our approach.
>
> We hope this explanation offers a more coherent insight into our methodology and addresses the queries raised.
>
> 3. Applicability to generative tasks:
>
> Response:
>
> We appreciate your feedback. As stated both in our abstract and throughout the paper, our proposed methodology specifically targets NLP classification tasks. The approach we have developed centers around integrating a personalized matrix directly into the model's classification layer. Given that generative tasks possess a fundamentally different problem formulation and output architecture compared to ours, they were not within the purview of our study.
>
> In our revised version, we will articulate this more clearly in the "limitations" section to ensure comprehensive coverage of the topic's scope and ensure there's no ambiguity regarding our focus.
>
> We trust this amendment provides clarity and addresses the nuances of our research.
>
> Questions For The Authors:
>
> Q: Is the backbone of the model frozen when producing the first personalized local model at time $P_{t_0}$?
>
> Response: No, we use $P_{t_0}$ to denote the stage the local model personalized using arbitrary method. In our example, full personalized training as we have not introduced the lightweight fine tuning yet. However in real-world case, it can be learned with different methods as long as the personalized model can perform better than generalized global model.
>
>
> Q: Did you explore any experimentation without training the global model on all user's data?
>
> Response: Yes we explored that this setting with directly fine tuning the model with user's data respectively. We report our result here for only fine tuning the BERT and RoBERTa in news recommendation, the AUC is 55.86/%$\pm$ 4.05/% and 52.44/% $\pm$ 12.48/% respectively. For hate speech detection, the F-1 score is 40.93\% $\pm$ \%12.41 and 42.51\%$\pm$ \%24.76.
>
> From above result, solely relying on fine-tuning without global training led to models that were inadequately personalized and exhibited considerable performance variation. For instance, while our method achieved an AUC of 65.64\% for news recommendation, the result for a model that was only fine-tuned stood at 52.44\%. This significant performance discrepancy was consistent across various settings.
> The rationale behind this discrepancy is that task-specific global training embeds a broader knowledge base regarding the tasks into the model. Subsequent personalized training can then adjust this generalized knowledge to cater to individual user distributions. However, in the absence of a substantial dataset for a specific user, accurately modeling such a user-specific distribution becomes immensely challenging. This is compounded by the fact that local devices have constraints in terms of data storage capacity and require periodic updates.
>
>
>
>
> Q: Why do you only select the top 5 users? Does this mean the global model is trained on the data for just those 5 users?
>
>
>
> Response:
> For the news recommendation task, our initial decision to select the top 5 users from the dataset was primarily influenced by the dataset's limitations. The dataset becomes exceedingly sparse when including users with fewer data points, which compromises the quality of personalization. Insufficient data may fail to capture nuances specific to individual user distributions. To effectively execute personalized training, it's important that we consider data robust enough to model these user-specific tendencies and preferences.
>
> Regarding your second question: We've ensured a distinct partitioning of data for the global training phase and the personalized training phase. Upon identifying our target users, in this case, the 5 selected users, we meticulously extracted their data from the main dataset, double-checking to prevent any overlap with the global training data set. This guarantees that the data pertaining to these users is isolated and does not intersect with the global training pool. This protocol was consistently followed for both the news recommendation and hate speech detection tasks.
>
> Q: It's surprising that Full would perform worse. Why is this?
> When you say it finetunes all parameters of the new model, I'm assuming this is for each user correct?
>
> Response:
> In the full model parameter tuning as the global model is trained on global distribution, for personalized tuning, as training error drops, the test error increases. The data collected is non-i.i.d and imbalance in amount compared with global training data set. In news recommendation, the most frequent user only hold 103 records for training and testing. It is much less than the data used for global training which is over 150k records. This can cause the overfitting for limited data with respect to large model.
>
> Q: Line 497 - 500: In which experimentation is the applicability to new unseen users explored?
>
> Response: We apologize for the oversight. As our users for personalization is drawn from global dataset and any records of them will not be seen on global training stage. we refer our ability of fast adapt to new users as the applicability to new unseen users. These unseen users can accumulate their behaviors in a short period then our model can use these information to adapt model when replacing the backbone.
>
> Q: What is the performance gap across time gaps? How did FaLA perform at timestep 0 compared to timestep 1?
>
> Response: Here we only consider the scenario of adapting one gap. Since the previous backbone can be personalization in arbitrary ways as along as it contains the information of user-specific distribution, for a new update, we only need to hold the prediction score from previous global model. Here for timestep 0 we consider it as time before we apply FaLa to update personalized model but the local model has also shown the personalized performance if trained using other personalized methods.
>
> We will add the specification of this question into our paper to avoid this confusion again.
>
>
>
>
>
> Once again, thank you for your feedback. We believe that these revisions substantially improve our paper, making it clearer and more informative for the readers. We hope the revised version addresses your concerns and meets the criteria for publication at EMNLP.
>
> Warm regards.

---

### Official Review · Reviewer_MvrZ · 2023-08-05

**Soundness:** 4

**Excitement:**

4: Strong: This paper deepens the understanding of some phenomenon or lowers the barriers to an existing research direction.

**Paper Topic And Main Contributions:**

The paper is about how to replace a backbone network on a user device with minimal computational costs. The question is important because simple download-and-replace is not enough. The model have to be adjusted somehow to better solve personalized downstream tasks.

Contributions:
* New approach to tune the model on the user device after the replacing of the backbone model. The approach outperforms the baselines on News recommendation and Hate speech detection tasks in both quality and computational efficiency (FLOPS).

**Questions For The Authors:**

Question A: Do I understand it right that in line 207 the input to the NewsEncoder is a *set* of news L, not a single news article? Furthermore, is it also NewsEncoder in the equation on line 221 (subsection Hate speech detection)?

Question B: Could you please explain the equation 5 a little better? (where does it come from)

**Reasons To Accept:**

Reasons To Accept:
* New approach to change a backbone model on device with low cost.
* The approach is validated on two NLP tasks: News recommendation and Hate speech detection.
* Extensive experiments with many models and datasets.
* Ablation study.
* Thorough "Limitations" section which, evidently, highlights some limitations of the current research but also, probably, shows some directions of future research in the field, and overall is a very good example of how to approach the writing of the "Limitations" section.

**Reasons To Reject:**

I see no solid reason to reject the paper.

**Reproducibility:**

4: Could mostly reproduce the results, but there may be some variation because of sample variance or minor variations in their interpretation of the protocol or method.

**Reviewer Confidence:**

2: Willing to defend my evaluation, but it is fairly likely that I missed some details, didn't understand some central points, or can't be sure about the novelty of the work.

**Typos Grammar Style And Presentation Improvements:**

* Line 35 (and many other places): Missing space between text and reference. For example, "for edge devices(Vucetic et al ...)"
* Line 57 (and many other places): Excess space before dot or comma. For example, "such as BERT , ..."
* Line 97: duplicate "consumes only"
* Lines 27, 164: "Flops, FLOPS". Different symbols for the same notion.
* Lines 232, 311: "L2, l2". Also, different symbols for the same things.
* Line 256: Dot instead of comma.
* Line 262 (eq 2). Should be "\diag" (mathrm font), not just "diag" (italic font).
* Line 289 (eq 3). Duplicate "=" sign.
* Line 296 (eq 4). Not quite understand the clause with Cos \in TopK.
* Line 307 (eq 5). Probably wrong indices for "X" letter.
* Line 318 (eq 6). The top vertical space between text and equation is bigger than the bottom space.
* Line 343: Wrong formatting of negative number "-3" (this is not a minus sign).
* Line 431: FLOPS (Floating point operations per second). I guess it is too late for such an explanation :)
* Line 494: "F-1" splits on two lines.
* Table 3: not all best values are in bold (there are two lines with no bold values at all).
* Table 4: The "%" sign in table rows can be omitted. Just name the column, eg. "Random Initial, %" and so on.
* Table 5: In my opinion, it would be better to span the values in Task column on several lines (so as to do without x5 repeating).
* Line 635: "Specifically, we preserve the predicted logits from old logits ..." The sentence probably needs revising.
* Line 648: "As our methods can efficiently bring personal initialization and bring efficient parameter tuning." Also, probably needs revising.
* Line 650: "folders" ?
* Table 6: Strange formatting: it takes two text columns, and is placed on a separate page.
* Figures 2 and 3: small font size.
* Line 854: Typo in "clusters"

---

> ### Author Rebuttal · Authors · 2023-08-29
>
> Dear Reviewer,
>
> We sincerely appreciate your thorough review and detailed feedback on our submission. Your insights have been invaluable in helping us improve the quality and clarity of our paper. Below are our responses to each of your comments:
>
> Reasons To Accept and Contributions:
>
> We are pleased that you recognize the value of our new approach and its potential implications in the field. We also acknowledge and appreciate your commendation on our "Limitations" section.
>
> Questions For The Authors:
>
> Question A: Do I understand it right that in line 207 the input to the NewsEncoder is a set of news L, not a single news article? Furthermore, is it also NewsEncoder in the equation on line 221?
>
> You're right about line 207; the input to the NewsEncoder is indeed a set of news
> $L$, not a single news article. We apologize for any confusion caused and will clarify this in the revised manuscript. Also, regarding line 221 in the Hate speech detection subsection, you are correct. We use the same denotation as in this task the comments are same encoded with news encoder like previous task into numeric space. We will ensure clarity in the equation's representation.
> Question B: Equation
> In the equation 5, it can be viewed as prediction layer of a classifier or similarity calculation equation.
> We would like to make our prediction which is the summation of $\theta X_i$ calculated by Hadamard product $\odot$ to be adjusted equal to the prediction of old model which is $\hat{Yt}_{i-1}$ using a parameter vector $\theta$. This parameter vector is the diagonal elements of previously mentioned matrix from this Equation 2. Equation 5 is reformulated from equation 2 to adapt to a form of linear optimization.
>  We understand that its presentation might be confusing, and will enhance the description in the revised version to offer a clearer understanding by explicitly explaining each element and operations in detail.
>
>  We believe these revisions have improved the readability and presentation of our paper. Once again, thank you for your constructive feedback. We hope the revised version addresses your concerns and meets the standards of EMNLP.
>
> Warm regards.

---

### Meta-Review · Area_Chair_4pio · 2023-09-17

**Recommendation:** 3

**Metareview:**

The paper is commended for addressing an interesting and important problem of language model backbone replacement for personalized downstream tasks in an on-device, non-stationary scenario. The clear separation between cloud and edge is acknowledged as crucial for the efficiency and privacy of ML models on low-powered devices, making this a relevant and timely topic. Reviewers appreciate that the proposed method achieves over 1000x computation reduction in FLOPs for back-propagation while outperforming comparable transfer learning methods. Additionally, the paper is recognized for conducting extensive experiments with various models and datasets, as well as including an ablation study.

However, reviewers raise concerns about the clarity and presentation of the paper. They note that the problem definition is not clear, and the paper lacks a clear, concise summary of the training procedure. There are concerns about the lack of certain experimental details. Reviewers suggest reporting memory usage for each method and providing actual running times in addition to FLOPs computation. I believe the inclusion of these details would strengthen the paper's experimental evidence.

Overall, the paper addresses an important and interesting problem with significant computational and performance improvements, but it needs to address concerns related to clarity, presentation and experimental details. Addressing these issues would enhance the paper's quality.

---

### Decision · Program_Chairs · 2023-10-07

**Decision:**

Accept-Findings

**Comment:**

The paper is commended for addressing an interesting and important problem of language model backbone replacement for personalized downstream tasks in an on-device, non-stationary scenario. The clear separation between cloud and edge is acknowledged as crucial for the efficiency and privacy of ML models on low-powered devices, making this a relevant and timely topic. Reviewers appreciate that the proposed method achieves over 1000x computation reduction in FLOPs for back-propagation while outperforming comparable transfer learning methods. Additionally, the paper is recognized for conducting extensive experiments with various models and datasets, as well as including an ablation study.

However, reviewers raise concerns about the clarity and presentation of the paper. They note that the problem definition is not clear, and the paper lacks a clear, concise summary of the training procedure. There are concerns about the lack of certain experimental details. Reviewers suggest reporting memory usage for each method and providing actual running times in addition to FLOPs computation. I believe the inclusion of these details would strengthen the paper's experimental evidence.

Overall, the paper addresses an important and interesting problem with significant computational and performance improvements, but it needs to address concerns related to clarity, presentation and experimental details. Addressing these issues would enhance the paper's quality.